# Copper-catalyzed asymmetric C(sp³)-H cyanoalkylation of glycine derivatives and peptides

Rupeng Qi[1,2,4], Qiao Chen[1,4], Liangyu Liu[3,4], Zijian Ma[3], Da Pan[1], Hongying Wang[3], Zhixuan Li[3], Chao Wang ✉[2,3] & Zhaoqing Xu ✉[2,3]

Alkylnitriles play important roles in many fields because of their unique electronic properties and structural characteristics. Incorporating cyanoalkyl with characteristic spectroscopy and reactivity properties into amino acids and peptides is of special interest for potential imaging and therapeutic purposes. Here, we report a copper-catalyzed asymmetric cyanoalkylation of C(sp³)-H. In the reactions, glycine derivatives can effectively couple with various cycloalkanone oxime esters with high enantioselectivities, and the reaction can be applied to the late-stage modification of peptides with good yields and excellent stereoselectivities, which is useful for modern peptide synthesis and drug discovery. The mechanistic studies show that the in situ formed copper complex by the coordination of glycine derivatives and chiral phosphine Cu catalyst can not only mediate the single electronic reduction of cycloalkanone oxime ester but also control the stereoselectivity of the cyanoalkylation reaction.

In recent years, the combination of transition-metal catalysis and radical chemistry has emerged as a powerful synthetic method for the construction of a variety of C-C and C-heteroatom bonds under mild conditions[1]. In particular, the transition metal-catalyzed radical cross-coupling allows the stereoselective construction of C(sp³)-C(sp³) bonds, which is often difficult to realize through the traditional ionic-based cross-coupling reactions[2]. Although significant progress has been achieved, the stereoselective alkylation via asymmetric radical cross-coupling of non-acidic C(sp³)-H bonds is still challenging and highly desired. This could attribute to the unavailability of p-orbitals of C(sp³)-H bonds to interact with the transition metal catalyst[3].

Alkylnitriles play important roles in many application fields due to their unique electronic and structural properties. A large number of pharmaceuticals, bioactive natural products and functional materials contain such structural motifs[4–9]. Meanwhile, the nitrile group is a versatile synthon as it can be readily transformed into a range of useful functional groups, such as carboxylic acids, amides, amines, aldehydes, ketones, etc.[10–14]. Ever since the pioneering study of Boivin et al.

in the 1990s, the iminyl radical-triggered C-C bond cleavage of cycloalkanone oximes has been successfully applied to synthesize a series of functionalized alkylnitriles under cyanide-free conditions (Fig. 1, eq. 1)[15–24]. Despite significant progress, the catalytic enantioselective variants of these reactions remain largely unexplored, and the reported reactions were restricted to the Giese additions of alkenes or the couplings with alkynes[25–28]. In contrast, the direct stereoselective cross-couplings of the key alkylnitrile radicals with other alkyl radicals to form chiral C(sp³)-C(sp³) bonds were still underdeveloped, especially with the C(sp³)-H as alkyl radical precursors.

Unnatural amino acids have encountered widespread applications for the preparation of biologically active molecules and peptidomimetic drugs[29,30]. Introducing functional groups with characteristic spectroscopy and reactivity properties (e.g., -N₃, -CN, and etc.) into amino acids and peptides is of special interest for potential imaging and therapeutic purposes[31–34]. Given the importance of cyano-containing unnatural amino acids in drug discovery and functional material development, the practical and stereoselective

[1]School of Pharmacy, Lanzhou University, 730000 Lanzhou, China. [2]Research Unit of Peptide Science, Chinese Academy of Medical Sciences, 2019RU066, 730000 Lanzhou, China. [3]Key Laboratory of Preclinical Study for New Drugs of Gansu Province, School of Basic Medical Sciences, Lanzhou University, 730000 Lanzhou, China. [4]These authors contributed equally: Rupeng Qi, Qiao Chen, Liangyu Liu. ✉e-mail: wangchao@lzu.edu.cn; zqxu@lzu.edu.cn

**Fig. 1 | Radical functionalization of glycine derivatives and peptides.** (1) The generation of alkylnitrile radicals. (2) Photo-induced Cu-catalyzed asymmetric C(sp³)-H alkylation. (3) Asymmetric C(sp³)-H cyanoalkylation of glycine derivatives and peptides. PC photocatalyst, TM transition metals.

synthetic method for constructing this type of compounds is highly desired.

In recent years, copper-catalyzed asymmetric C(sp³)-H functionalization through radical processes has attracted extensive attention[35–39]. We recently developed a photo-induced and Cu-catalyzed asymmetric C(sp³)-H alkylation of glycine derivatives for the synthesis of unnatural amino acids[40]. The mechanistic studies revealed that the photo irradiation was essentially required to generate a high-valent organometallic copper (Cu^III) intermediate, which was crucial to converting the glycine C(sp³)-H to the corresponding alkyl radical via single electron transfer (SET) reduction and deprotonation sequence (Fig. 1, eq. 2). Interestingly, Gong recently developed a Fe promoted glycine C(sp³)-H cyanoalkylation, in which Fe^III served as the SET oxidant to generate the glycine N-radical[41]. Meanwhile, Zhou et al.[42], Zuo et al.[28] and Zhang et al.[43] independently disclosed that the redox-active cyclobutanone oxime could forge the formation of Cu^III species. Inspired by these reports and on the basis of our long-standing interest in Cu-catalyzed radical couplings[40,44–50], we hypothesized that trapping of the alkylnitrile radical following ring-opening of cyclobutanone oxime by a Cu complex would in situ form a Cu^III complex, which would possibly enable glycine C(sp³)-H to form an alkyl radical under basic conditions. Moreover, with the assistance of a chiral ligand, the asymmetric C(sp³)-H cyanoalkylation might be achieved as a practical complementary approach to the current photo-irradiation strategy[40,50]. Importantly, the achievement of this reaction would provide a useful tool for preparing cyanoalkylated unnatural α-amino acids and late-stage modification of peptides (Fig. 1, eq. 3).

## Results

### Investigation of reaction conditions

In this program, we first examined the capability of the chiral Cu complex for catalyzing the redox process and its ability to control the stereoselectivity of radical cyanoalkylation using ethyl (4-methoxyphenyl)glycinate **1a** and cyclobutanone oxime ester **2a** as model substrates. Asymmetric cyanoalkylation was tested employing different chiral ligands (**L1–L7** in Table 1) using DABCO as the base and acetone as the solvent, whereas no stereoselectivities were observed in all cases (Fig. 2, eq. 1). We assume that the intermolecular coordination between the chiral catalyst and alkyl radical is quite weak and random, and it is difficult to control the stereochemistry of competitive racemic background reaction. Thus, enhancing the interaction between the

chiral catalyst and substrate is an effective strategy to realize the stereoselective cyanoalkylation of C(sp³)-H[40].

Therefore, we embed the easily removed N donor unit into the substrate, by which the chiral catalyst and substrate would jointly constitute a new chiral catalytic system. The results show that electron transfer (or charge transfer) and stereocontrol can be defined within one molecule, and the asymmetric cyanoalkylation of glycine derivatives was achieved through the stereoselective reductive elimination of Cu^III species (Fig. 2, eq. 2). Using tert-butyl (5-methoxyquinolin-8-yl) glycinate **1b** and cyclobutanone oxime ester **2a** as model substrates, to our delight when Cu(MeCN)₄PF₆ (10 mol%) and (S)-Phanephos (**L1**, 11 mol%) were employed, the ee of C(sp³)-H cyanoalkylation product was dramatically increased to 80%.

After exploring various chiral ligands, basic additives, Cu catalysts, solvents, and other parameters (see Supplementary Information for details), the optimized reaction conditions were determined (Table 1). The desired product **3ba** was obtained in 80% yield and 93% ee at 0 °C for 18 h in the presence of Cu(MeCN)₄PF₆ (10 mol%), (S)-An-Phanephos (15 mol%), DABCO (2.0 equiv), and acetone (0.2 M) (Table 1, entry 1). Control experiments indicated that copper salt and basic additive are indispensable for the reaction (entries 2–3). Furthermore, Cu(MeCN)₄PF₆ also can independently catalyze the C(sp³)-H cyanoalkylation of glycine derivative in the absence of (S)-An-Phanephos, leading to the formation of the racemic product in 28% yield (entry 4). Notably, the efficiency of cyanoalkylation was not improved under LED irradiation (entry 5). The coupling reaction was sensitive to oxygen, and the yield dropped sharply to 36% under air conditions (entry 6).

### Substrate scopes

With the optimized reaction conditions in hand, we explored the substrate scope of oxime esters. As summarized in Fig. 3, the catalytic system exhibited a broad substrate scope and good functional group tolerance. It could accommodate cyclobutanone, 3-oxetanone, Cbz and Boc protected azetidinone derived oxime esters (**2a-d**), delivering products **3ba-bd** in satisfactory yields and enantiomeric excesses. X-ray crystallographic analysis of product **3ba** confirmed the (S)-configuration of the newly formed stereochemistry. Meanwhile, (**R**)-**3ba** was obtained in 70% yield and −90% ee when (R)-An-Phanephos was used instead of (S)-An-Phanephos. The reaction of disubstituted oxime esters **2e** and **2 f** also proceeded smoothly to afford the desired products **3be** and **3bf** in good yields and ees, respectively. In addition, the

## Table 1 | Optimization of the reaction conditions

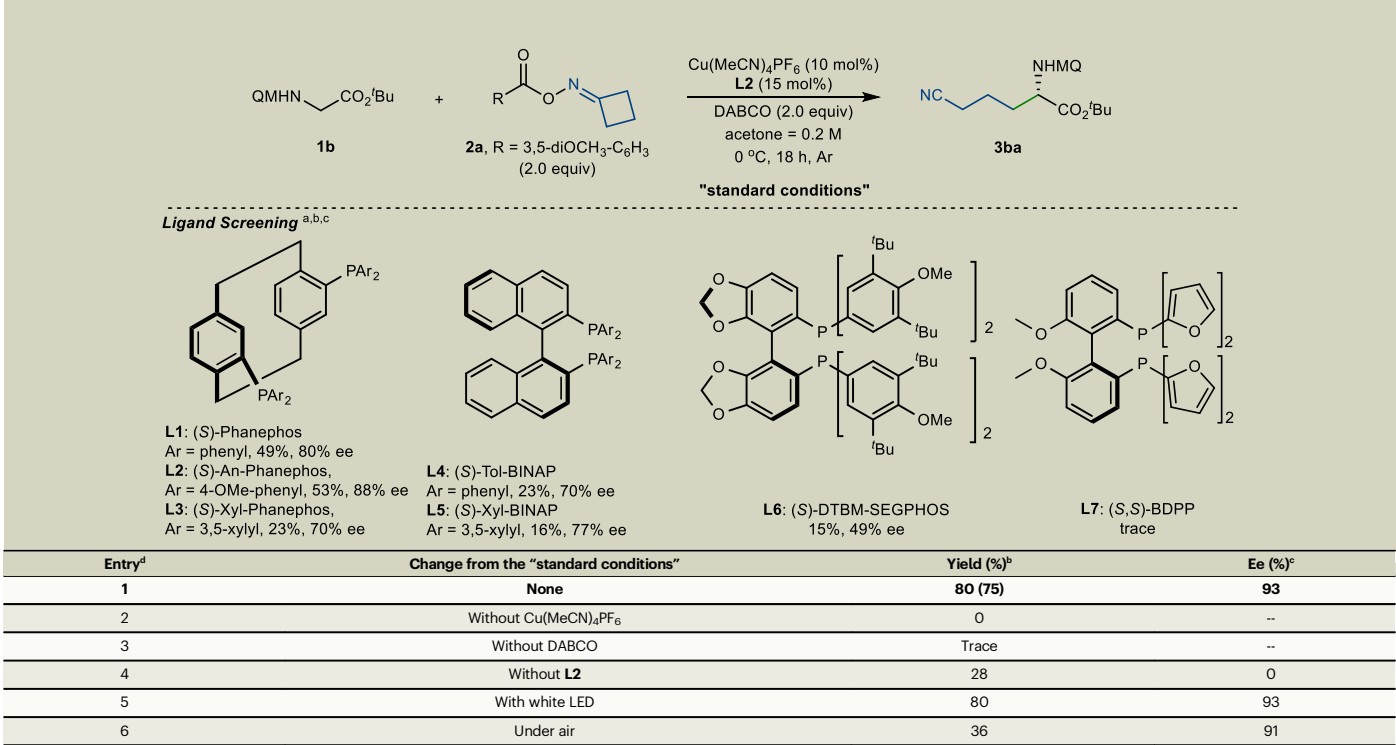

| Entry[d] | Change from the "standard conditions" | Yield (%)[b] | Ee (%)[c] |
|---|---|---|---|
| **1** | **None** | **80 (75)** | **93** |
| 2 | Without Cu(MeCN)$_4$PF$_6$ | 0 | -- |
| 3 | Without DABCO | Trace | -- |
| 4 | Without **L2** | 28 | 0 |
| 5 | With white LED | 80 | 93 |
| 6 | Under air | 36 | 91 |

[a]Condition: **1b** (0.05 mmol), **2a** (1.5 equiv), Cu(MeCN)$_4$PF$_6$ (10 mol%), **L2** (11 mol%), DABCO (2.0 equiv), acetone (1.0 mL), room temperature, 12 h, and under argon atmosphere.
[b]Yield was determined by $^1$H NMR using 4-bromobenzaldehyde as an internal standard and isolated yield in parentheses.
[c]Ee (enantiomeric excess) was determined by HPLC on a Chiralpak IA-H column.
[d]Standard conditions: **1b** (0.1 mmol), **2a** (2.0 equiv), Cu(MeCN)$_4$PF$_6$ (10 mol%), **L2** (15 mol%), DABCO (2.0 equiv), acetone (0.5 mL), 0 °C, 18 h, and at argon atmosphere.

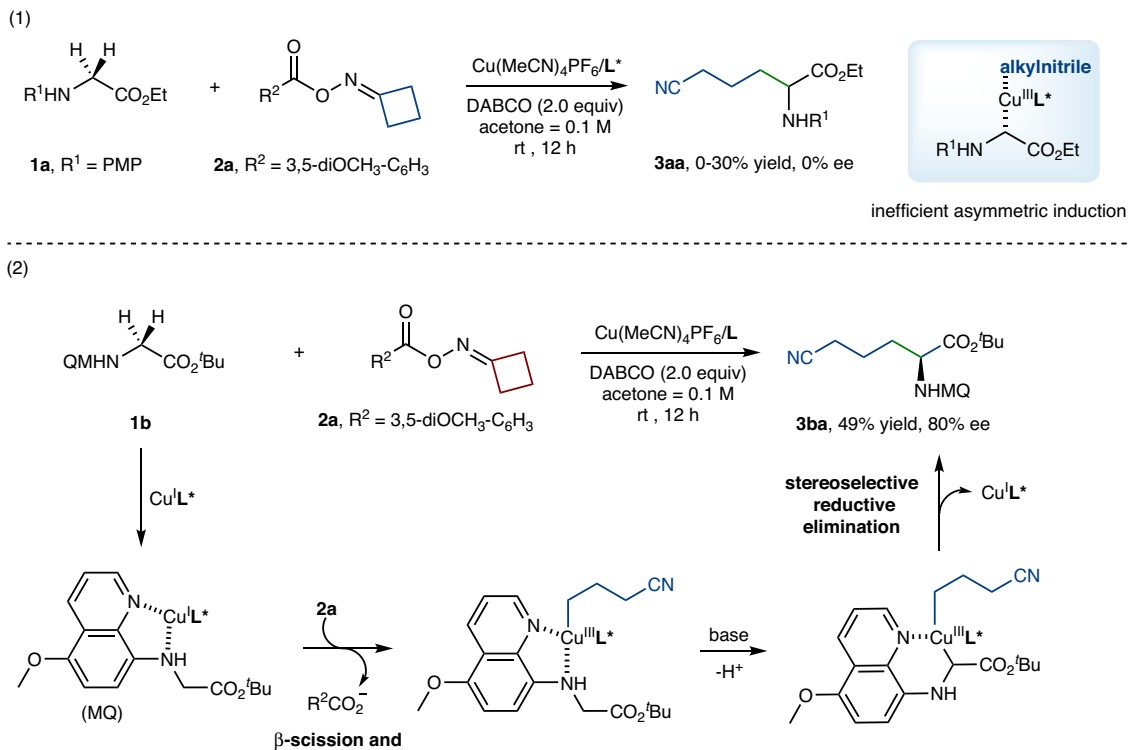

**Fig. 2 | Copper-catalyzed asymmetric C(sp³)-H cyanoalkylation of glycine derivatives and peptides.** (1) Asymmetric C(sp³)-H cyanoalkylation of ethyl (4-methoxyphenyl)glycinate. (2) Asymmetric C(sp³)-H cyanoalkylation of *tert*-butyl (5-methoxyquinolin-8-yl)glycinate. PMP 4 -methoxyphenyl.

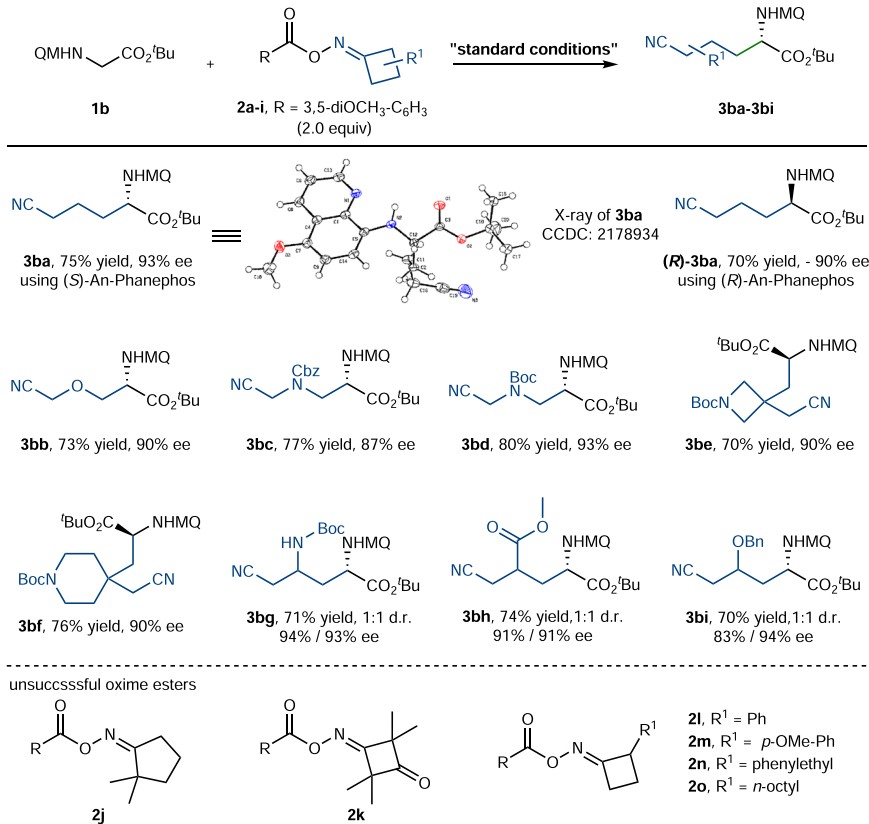

**Fig. 3 | Substrate scope with respect to oxime esters.** Standard conditions: **1b** (0.1 mmol), **2** (2.0 equiv), Cu(MeCN)₄PF₆ (10 mol%), **L2** (15 mol%), DABCO (2.0 equiv), acetone (0.5 mL), 0 °C, 18 h, argon atmosphere. Isolated yields based on **1b** after chromatographic purification. Ee was determined by chiral HPLC. D.r. (diastereo ratio) was determined by ¹H NMR analysis.

monosubstituted oxime esters **2g-i** with -NHBoc, ester and ether groups at the 3-position were also well tolerated, with the corresponding products **3bg-bi** isolated in 70–74% yields and 83–94% ees. As a limitation of this protocol, the secondary and tertiary cyanoalkyl radicals derived from **2l-2o** were less efficiently captured by the copper catalysts compared to primary cyanoalkyl radicals and gave negative results.

Further expansion of the substrate scope was focused on glycine derivatives (Fig. 4). Glycine amide (**1c**), glycine ester (**1d**) and glycine derivative bearing different *N*-aryl groups (**1e**) were well tolerated (**3ca-ea**, 65–77% yields). Furthermore, a variety of dipeptides (Gly-Val, Gly-Pro, Gly-Glu, Gly-Phe, Gly-Met, Gly-Thr and Gly-Lys) substrates were prepared to test the functional group tolerance of this reaction with various amino acid residues incorporated. Gratifyingly, the corresponding cyanoalkylated peptides **3fa-la** afforded good yields and excellent stereoselectivities with other amino acid residues untouched (Fig. 4A).

Having established proof-of-concept with the above results, we became interested if our reactions could be applied in C(sp³)-H cyanoalkylation of polypeptides (Fig. 4B). Gratifyingly, the late-stage C(sp³)-H cyanoalkylation of pentapeptide (Gly-Leu-Phe-Ser-Lys) and hexapeptide (Gly-Leu-Tyr-Ser-Phe-Ala) derived substrates reacted smoothly and gave the corresponding cyanoalkylation products in good yields (**3ma**, 63%; **3na**, 60%) and high diastereoselectivities (>20:1). It was worth noting that the reactions between hexapeptide (Gly-Leu-Tyr-Ser-Phe-Ala and Gly-Leu-Phe-Gly-D-Thr-Tyr) substrate and different oxime esters (**2a, 2b, 2e**, and **2f**) were achieved in uniformly good yields and high diastereoselectivities, which further highlighted the generality of this method in modification of complex molecules.

## Synthetic applications

To further illustrate the application potential of the product, several transformations were conducted (Fig. 5). Firstly, the deprotection of the cyanoalkylation product **3ba** proceeded smoothly under simple procedures in high yield and did not erode the ee (Fig. 5, eq. 1). Secondly, treatment of the **3ba** with Raney-Ni and H₂ in pyridine/EtOH/H₂O afforded the corresponding amine **5** in 70% yield and 90% ee (Fig. 5, eq. 2). Furthermore, the cyano group of **3ba** was easily converted to amide **6** in the presence of Pd catalyst (Fig. 5, eq. 3).

## Mechanistic studies

In order to gain some insight into the mechanism, several radical trapping experiments were carried out (Fig. 6). The radical trapping experiments with TEMPO (2,2,6,6-tetramethylpiperidin-1-oxyl, completely suppressed) or BHT (2,6-ditert-butyl-4-methylphenol, partially suppressed) indicated that a radical pathway was involved in this transformation. Meanwhile, radical trapping product **7** was isolated in 30% yield in the presence of TEMPO, which suggested the formation of cyanoalkyl radical in the reaction system (see Supplementary Information for details). Furthermore, the HRMS analysis of the original reaction mixture detected the formation of glycinate homo-coupling product **8** when the dosage of **2a** was 1.0 equiv (Fig. 6, eq. 1). Moreover, when the reactions of **1b** and **2a** were carried out under air, oxidized glycine derivative **9** was obtained in 40% yield (Fig. 6, eq. 2). These results indicated that glycinate radical was generated under standard reaction conditions. *N*-CH₃-substituted glycinate **1p** did not give any cyanoalkylation product with quantitive recycling of the starting material, indicating that a single free hydrogen atom on N atom is crucial for this reaction (Fig. 6, eq. 3). Notably, imine **1q** failed to give the cyanoalkylation product **3da**, revealing the Cu-mediated SET

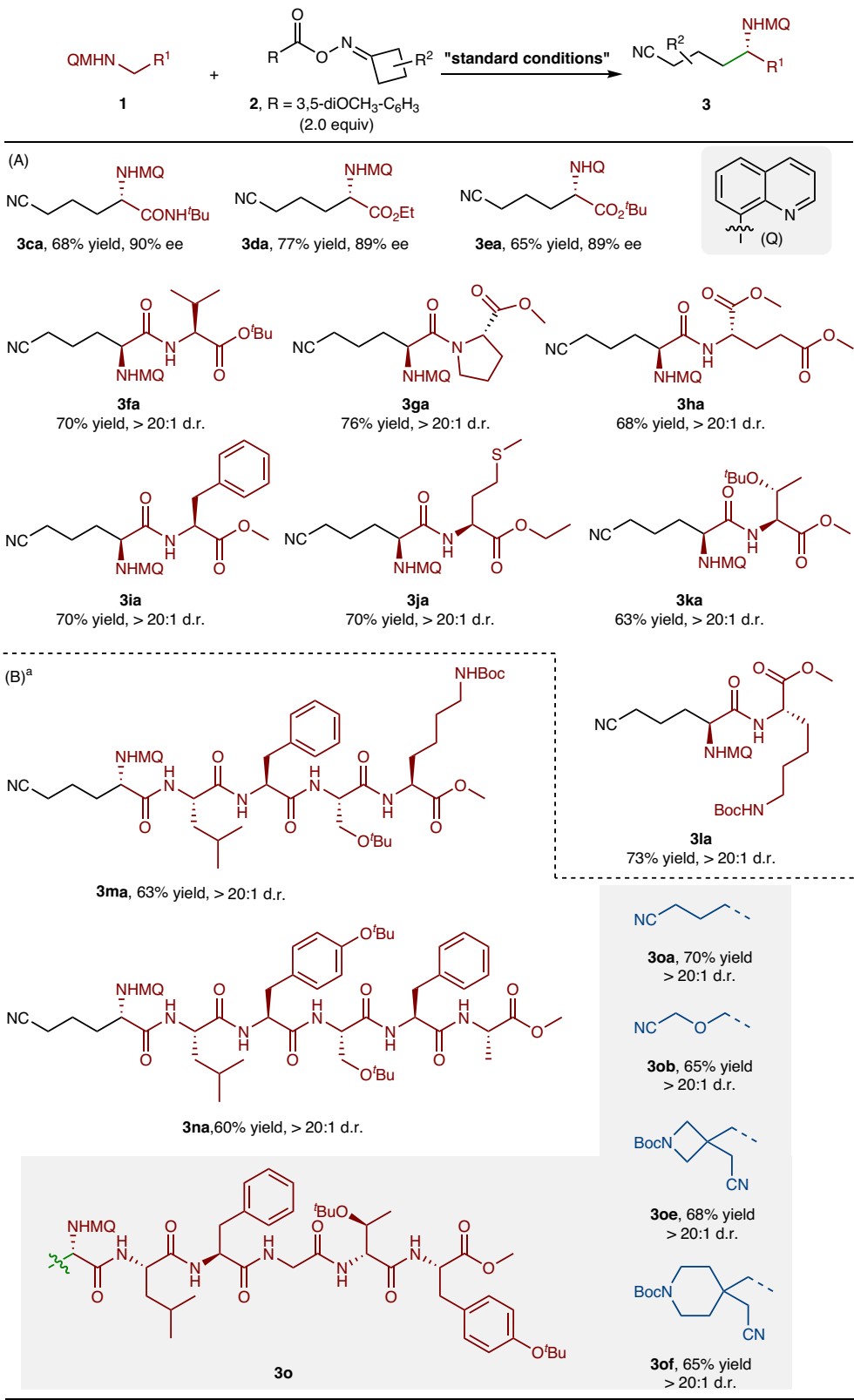

**Fig. 4 | Substrate scope with respect to glycine derivatives and peptides.**
Standard conditions: **1** (0.1 mmol), **2** (2.0 equiv), Cu(MeCN)₄PF₆ (10 mol%), **L2** (15 mol%), DABCO (2.0 equiv), acetone (0.5 mL), 0 °C, 18 h, under Ar. Isolated yields based on **1** after chromatographic purification. Ee was determined by chiral HPLC.

D.r. was determined by ¹H NMR analysis. ᵃDMF (0.5 mL) as solvent, 36 h. **A** Substrate scope with respect to glycine derivatives and dipeptides. **B** Substrate scope with respect to peptides.

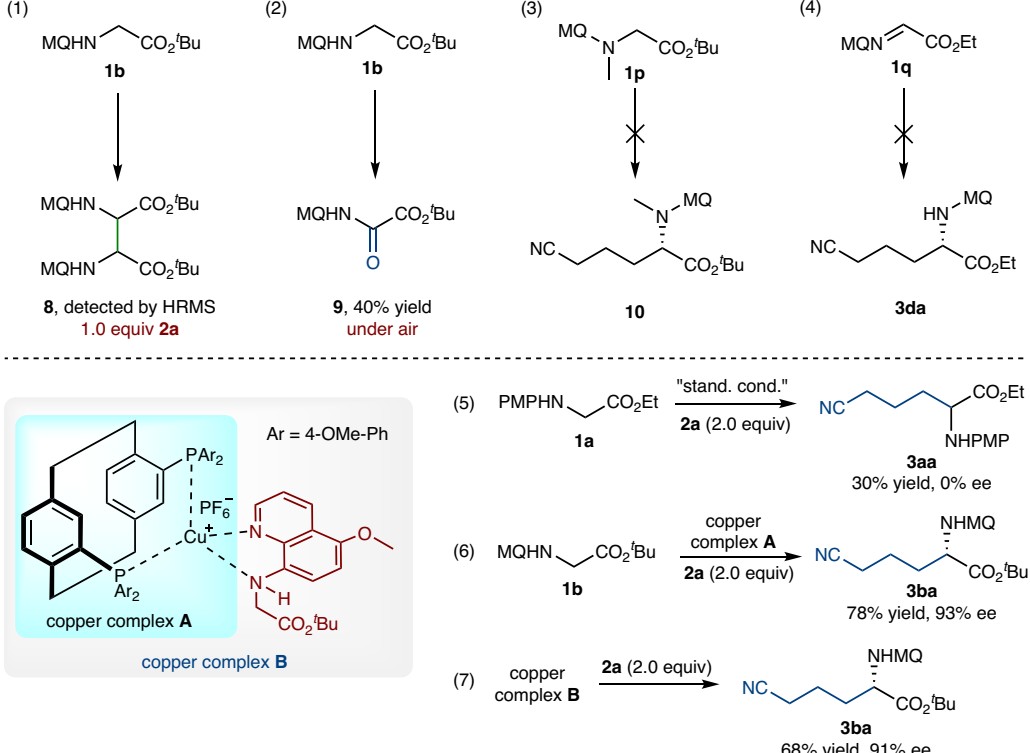

**Fig. 5 | Synthetic applications.** (1) Deprotection of **3ba**. (2) Conversion of cyanide to amine. (3) Conversion of cyanide to amide. CAN Ce(NH₄)₂(NO₃)₆, Py pyridine.

**Fig. 6 | Mechanistic studies.** (1) The formation of glycinate homo-coupling product. (2) The α-oxidation of glycinate. (3) The reaction with *N*-CH₃-substituted glycinate. (4) The reaction with imine. (5) The reaction with ethyl-(4-methoxyphenyl) glycinate. (6) The reaction with copper complex **A**. (7) The reaction with copper complex **B**.

pathway likely involved rather than the two-electron oxidation process (Fig. 6, eq. 4).

We prepared ((*S*)-An-Phanephos)Cu(CH₃CN)₄PF₆ complex **A** and copper complex **B** (combined complex **A** and **1b**) according to the literature procedure[51]. The ground state redox potentials of **2a** ($E_{redox}$ = −2.11 V vs. SCE in CH₃CN) and copper complex **B** ($E_{redox}$ = −2.38 V vs. SCE in CH₃CN) demonstrated that the SET oxidation of **2a** by copper complex **B** was feasible (see Supplementary Information for details). Some control experiments were also carried out, and the results are shown in Fig. 6 (eqs. 5–7). Ethyl (4-methoxyphenyl) glycinate **1a** could provide a racemic product under standard conditions (**3aa**, 30% yield, 0% ee, Fig. 6, eq. 5). The result revealed the importance of the coordination with 5-methoxyquinoline group for the chirality induction. When the

isolated Cu complex **A** was used as the catalyst, the corresponding cyanoalkylation product was obtained in 78% yield and 93% ee (Fig. 6, eq. 6). Furthermore, the isolated Cu complex **B** (1.0 equiv) was directly used as the substrate without any other catalyst in the system, the coupling product was obtained in 68% yield and 91% ee (Fig. 6, eq. 7). These results indicated that complex **B** likely formed in the reaction and act as the active catalytic species.

**Proposed mechanism**

Based on the above mechanistic studies and previous reports[52–55], a plausible mechanism was proposed in Fig. 7. The 5-methoxyquinolinyl-8-glycinate ester **1b** coordinates to the Cu¹L* and in situ forms a chiral Cu(I) intermediate **A**. Then, a SET reduction of oxime ester **2a** by intermediate **A** occurs, followed

**Fig. 7 | Proposed mechanism.** Copper-catalyzed asymmetric C(sp³)-H cyanoalkylation of glycine derivatives.

by fragmentation to afford cyclic iminyl radical **2a-A** and oxidized Cu(II) intermediate **B**. Next, cyclic iminyl radical **2a-A** undergoes regioselective ring-opening C-C bond cleavage to form cyanoalkyl radical **2a-B**. At this point, cyanoalkyl radical **2a-B** was captured by Cu(II) intermediate **B** to produce the high-valent Cu(III) intermediate **C**. Then, the copper-mediated intramolecular oxidation of the N atom produces Cu(II) intermediate **D**, and DABCO-promoted deprotonation gives radical intermediate **E**, which subsequently attacks the copper center to form a chiral Cu(III) intermediate **F**. Finally, the stereoselectively reductive elimination delivered the desired product **3ba**.

## Discussion

In conclusion, we report a copper-catalyzed asymmetric C(sp³)-H cyanoalkylation of glycine derivatives and peptides. The reactions feature mild conditions, excellent enantioselectivity and broad substrate scope. Given the significance of introducing cyanoalkylation into amino acids and peptides for potential imaging and therapeutic purposes, we predict that our asymmetric C(sp³)-H cyanoalkylation would provide new approaches to the synthesis of unnatural α-amino acids and late-stage functionalization of bioactive compounds, and would be useful for modern peptide synthesis and drug discovery.

## Methods

### General procedure A (3ba-bi, 3ca-la)

To an oven-dried 10-mL quartz test tube with a stirring bar, derivatives of glycine (0.1 mmol) were added, followed by the addition of Cu(MeCN)₄PF₆ (0.01 mmol, 3.7 mg) and (*S*)-An-

Phanephos or (*R*)-An-Phanephos (0.015 mmol, 10.5 mg). Then, the air was withdrawn and backfilled with Ar (three times). Acetone (0.25 mL) was added, and the mixture was stirred at room temperature for 40 min. Subsequently, oxime esters (0.2 mmol) and DABCO (0.2 mmol, 22.4 mg) dissolved in acetone (0.25 mL) were added to the abovementioned mixed solution by syringe. Thereafter, the test tube was transferred to a low-temperature device, where it was reacted for 18 h at 0 °C. Then, the reaction was quenched with water (1 mL), extracted with ethyl acetate (3 × 1.5 ml), dried over anhydrous sodium sulfate, concentrated in vacuo, and purified by column chromatography (hexane/ethyl acetate) to give the product.

### General procedure B (3ma-oa, 3ob, 3oe, and 3of)

To an oven-dried 10-mL quartz test tube with a stirring bar, peptide substrates (0.1 mmol) were added, followed by the addition of Cu(MeCN)₄PF₆ (0.01 mmol, 3.7 mg) and (*S*)-An-Phanephos (0.015 mmol, 10.5 mg). Then, the air was withdrawn and backfilled with Ar (three times). DMF (0.25 mL) was added, and the mixture was stirred at room temperature for 40 min. Subsequently, oxime esters (0.2 mmol) and DABCO (0.2 mmol, 22.4 mg) dissolved in DMF (0.25 mL) were added to the abovementioned mixed solution by syringe. Thereafter, the test tube was transferred to a low-temperature device, where it was reacted for 36 h at 0 °C. Then, the reaction was quenched with water (1 mL), extracted with ethyl acetate (3 × 1.5 ml), dried over anhydrous sodium sulfate, concentrated in vacuo, and purified by column chromatography (hexane/ethyl acetate or dichloromethane/methanol) to give the product.

## Data availability

The authors declare that the data supporting the findings of this study, including experimental details and compound characterization, are available within the article and its supplementary information file and all other data are available from the respective authors upon request. The X-ray crystallographic coordinates for structures reported in this study have been deposited at the Cambridge Crystallographic Data Centre (CCDC) under deposition number 2178934. These data can be obtained free of charge from The Cambridge Crystallographic Data Centre via https://www.ccdc.cam.ac.uk/.

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

## Acknowledgements

Supported by the National Natural Science Foundation of China (21971098 and 22271126 for Z.X., 22201112 for C.W.), Innovation Project of Medicine and Health Science and Technology of Chinese Academy of Medical Sciences (2019-I2M-5-074 for Z.X.), Key R&D Project of Gansu Province (22YF7WA010 for Z.X.), and Baiyin Science and Technology Planning Project (2022-2-28G for C.W.).

## Author contributions

Z.X. and C.W. conceived the idea, directed the project and designed the experiments; R.Q., Q.C., L.L., Z.M., D.P., H.W. and Z.L. performed the experiments; R.Q. and Q.C. analyzed the data; C.W. and Z.X. wrote the manuscript.

## Competing interests

The authors declare no competing interests.
