## [Peer Review File · Nature Communications]

Copper-catalyzed asymmetric C(sp³)-H cyanoalkylation of glycine derivatives and peptidesREVIEWER COMMENTS

Reviewer #1 (Remarks to the Author):

Xu, Wang and co-workers reported a copper catalyzed asymmetric cyanoalkylation via quinoline-directed C(sp³)-H functionalization strategy. The authors made much effort to expand the substrate scope regarding structurally different cycloalkanone oxime esters and quinoline-derived α -amidoacetates or peptides with good yield and excellent stereoselectivities. The chemistry is useful to modify small peptides. However, this strategy is the continuation of their previous works (J. Am. Chem. Soc. 2021, 143, 12777; Angew. Chem. Int. Ed. 2022, 61, e202200822). Almost same catalytic system and combined radical strategy using cycloalkanone oxime esters as radical donors make this method less attractive. The authors emphasized on photo free conditions, why? As compared with previous works, is it the highlight? The authors used quinoline-derived glycine ester, rather than glycine, which would make readers misunderstand. The peaks are very small in most of spectrums. The purity of products is questionable. This reviewer thinks this work is not suitable for Nature Communications due to it does not meet the requirement of original research.

Reviewer #2 (Remarks to the Author):

In this manuscript, Xu and coworkers report a copper-catalyzed cyanoalkylation of C(sp³)-H bonds and this reaction provide a useful tool for preparing cyanoalkylated unnatural α -amino acids and late-stage modification of peptides. Although an auxiliary group is needed to facilitate the reaction, the protecting group can be easily removed. Overall, I like this chemistry, and this manuscript may be publishable once the following items have been addressed.

1. Alkyl-alkyl bonds in line 4 of Introduction is suggested to be revised to "C(sp³)-C(sp³) bonds"
2. Radical symbol in intermediate E (Fig 2 and Fig 7) should be removed and a bond between Cu and carbon radical center should be added. Otherwise, the valent of Cu is misleading.
3. From the proposed mechanism, the N-H (substrate 1) is not essential to the success of reaction, the authors should try some substrates where N-H bond was substituted by other groups such as Me, Bn, Boc etc.
4. Some of the data in Fig 3 are inconsistent with those in support information, such as 3bf. The authors should carefully proofread the data and correct the typos.

Reviewer #3 (Remarks to the Author):

This manuscript by Xu et al. described an asymmetric protocol of phosphine-based copper catalysis enabling direct cyanoalkylation of glycine and peptides. Of essence, this interesting work has been relied on the tactic of single electron transfer events allowing to forge C(sp³)-C(sp³) bond at the catalytically active Cu metal center. Upon establishment of the optimal reaction conditions, a variety of cyclobutanone oxime esters have been demonstrated to be feasible in the cyanoalkylations of glycine derived substrate. In these cases, functional groups including NCbz, NBoc, ester and ethers were compatible. Regarding the scope of oligomeric peptides, moderate to good yields with excellent diastereoselectivities were attained, highlighting the great promise of this methodology for the syntheses of unnatural amino acids and peptides. Finally, a set of experimental studies were performed to elucidate the putative mechanism.

In the mind of Reviewer, this methodology is considered to be the extension to the previous reported by the same group (J. Am. Chem. Soc. 2021, 143, 12777). Along with extensive application of cyclobutanone oxime ester in the transition-metal catalyzed cyanoalkylation, this work represents as one of few successful asymmetric examples (for example, J. Am. Chem. Soc. 2021, 143, 13382). Together with the aforementioned results, Reviewer does agree that, this work is novel and potential. On the other hand, Reviewer found a few important points that should be clarified and addressed, then this article may be published in Nature Communications after major revision.

(1) The term “photo-free” described in the main text is inappropriate and should be removed. There is nonsense to emphasize the reaction conditions without the need of light irradiation. In Fig. 1b, authors should properly show the previous catalysis with N-hydroxyphthalimide esters, highlighting different sources of alkyl radicals.

(2) The current applications of oxime esters are relatively narrow. Additional variants derived from alpha-monosubstituted cyclobutanones should be examined to showcase the regioselectivities of the forming alkyl radicals as well as the diastereoselectivities of the case generating two adjacent chiral centers.

(3) Table 1 of optimization studies should be reformatted. Concise results from the uses of the drawn ligands should be included.

(4) Data of ³¹P-NMR for copper complex A and B should be included in supporting information.

(5) In Fig. 7, please check carefully the structures of A and B. In addition, investigation of the specific LMCT step (C → D) is too preliminary to state the event of single electron transfer from N to Cu(III). How would the electron density of the quinoline ring affect this SET event as well as the

performance of catalysis? And more information about the subsequent deprotonation is highly recommended by the measurement of KIE.

REVIEWERS' COMMENTS:

Reviewer #1:

Comments: Xu, Wang and co-workers reported a copper catalyzed asymmetric cyanoalkylation via quinoline-directed C(sp³)-H functionalization strategy. The authors made much effort to expand the substrate scope regarding structurally different cycloalkanone oxime esters and quinoline-derived -amidoacetates or peptides with good yield and excellent stereoselectivities. The chemistry is useful to modify small peptides. However, this strategy is the continuation of their previous works (J. Am. Chem. Soc. 2021, 143, 12777; Angew. Chem. Int. Ed. 2022, 61, e202200822). Almost same catalytic system and combined radical strategy using cycloalkanone oxime esters as radical donors make this method less attractive.

(1) The authors emphasized on photo free conditions, why? As compared with previous works, is it the highlight? The authors used quinoline-derived glycine ester, rather than glycine, which would make readers misunderstand.

Reply: We are grateful for the reviewer's comments to our work. Actually, the reaction pathway in this report is different to our previous work (JACS, 2021, 143, 12777; ACIE, 2022, 61, e202200822). In our previous work, the photo irradiation was crucial for the formation of the C-centred radical of glycine derivatives, which underwent a photo induced oxidative quenching step and a photo promoted intramolecular LMCT process. In the current work, we found that the C-centred radical of glycine derivatives can be generated in ground state, and therefore, it is obvious different to our reported works. Furthermore, we highly recognized the reviewer's comment and amended the term "glycine" to "glycine derivative" in the title to avoid misunderstanding.

Comments: (2) The peaks are very small in most of spectrums. The purity of products is questionable. This reviewer thinks this work is not suitable for Nature Communications due to it does not meet the requirement of original research.

Reply: We are grateful for the reviewer's comments to our work. According to the reviewer's suggestion, we adjusted the peak height of all the spectra, and the purity of the product was further improved.

Reviewer #2:

Comments: In this manuscript, Xu and coworkers report a copper-catalyzed cyanoalkylation of C(sp³)-H bonds and this reaction provide a useful tool for preparing cyanoalkylated unnatural α -amino acids and late-stage modification of peptides. Although an auxiliary group is needed to facilitate the reaction, the protecting group can be easily removed. Overall, I like this chemistry, and this manuscript may be publishable once the following items have been addressed.

(1) Alkyl-alkyl bonds in line 4 of introduction is suggested to be revised to "C(sp³)-C(sp³) bonds".

Reply: We are grateful for the reviewer's comments to our work. "Alkyl-alkyl bonds" has been modified to "C(sp³)-C(sp³) bonds" in the revised manuscript.

Comments: (2) Radical symbol in intermediate E (Fig 2 and Fig 7) should be removed and a bond between Cu and carbon radical center should be added. Otherwise, the valent of Cu is misleading.

Reply: We are grateful for the reviewer's comments to our work. The corresponding positions in Fig. 2 and Fig. 7 have been modified.

Comments: (3) From the proposed mechanism, the N-H (substrate 1) is not essential to the success of reaction, the authors should try some substrates where N-H bond was substituted by other groups such as Me, Bn, Boc etc.

Reply: We are grateful for the reviewer's comments to our work. In our mechanistic studies, the reaction of *N*-CH₃ substrate (Fig. 6C) was failed to provide cyanoalkylation product **10**. According to the reviewer's suggestion, we recently prepared the *N*-Bn substrate and used in our study. However, the corresponding cyanoalkylated product was not formed under standard reaction conditions. During the revision time, we have tried several methods to synthesize the *N*-Boc substrate, however, all the attempt were

failed. Above mentioned results suggested that the N-H is crucial to the success of the reaction.

Comments: (4) Some of the data in Fig 3 are inconsistent with those in support information, such as 3bf. The authors should carefully proofread the data and correct the typos.

Reply: We are grateful for the reviewer's suggestion. The corresponding mistakes were corrected (SI, p25), and we have went through the whole manuscript during the revision.

Reviewer #3:

Comments: This manuscript by Xu et al. described an asymmetric protocol of phosphine-based copper catalysis enabling direct cyanoalkylation of glycine and peptides. Of essence, this interesting work has been relied on the tactic of single electron transfer events allowing to forge C(sp³)-C(sp³) bond at the catalytically active Cu metal center. Upon establishment of the optimal reaction conditions, a variety of cyclobutanone oxime esters have been demonstrated to be feasible in the cyanoalkylations of glycine derived substrate. In these cases, functional groups including NCbz, NBoc, ester and ethers were compatible. Regarding the scope of oligomeric peptides, moderate to good yields with excellent diastereoselectivities were attained, highlighting the great promise of this methodology for the syntheses of unnatural amino acids and peptides. Finally, a set of experimental studies were performed to elucidate the putative mechanism.

In the mind of Reviewer, this methodology is considered to be the extension to the previous reported by the same group (J. Am. Chem. Soc. 2021, 143, 12777). Along with extensive application of cyclobutanone oxime ester in the transition-metal catalyzed cyanoalkylation, this work represents as one of few successful asymmetric examples (for example, J. Am. Chem. Soc. 2021, 143, 13382). Together with the aforementioned results, Reviewer does agree that, this work is novel and potential. On the other hand, Reviewer found a few important points that should be clarified and

addressed, then this article may be published in Nature Communications after major revision.

(1) The term “photo-free” described in the main text is inappropriate and should be removed. There is nonsense to emphasize the reaction conditions without the need of light irradiation. In Fig. 1b, authors should properly show the previous catalysis with N-hydroxyphthalimide esters, highlighting different sources of alkyl radicals.

Reply: We are grateful for the reviewer’s comments to our work. According to the reviewer’s suggestion, we have rewritten the “Introduction” section and adjusted our statements. Furthermore, we also showed the previous reactions with N-hydroxyphthalimide esters in Figure 1b.

Comments: (2) The current applications of oxime esters are relatively narrow. Additional variants derived from alpha-monosubstituted cyclobutanones should be examined to showcase the regioselectivities of the forming alkyl radicals as well as the diastereoselectivities of the case generating two adjacent chiral centers.

Reply: We are grateful for the reviewer’s suggestions. We recently synthesized four α -monosubstituted cyclobutanones (SI, page 4-6) and applied to the reaction. However, no desired products were detected under standard reaction conditions. Similar to the cases of alpha-disubstituted cyclobutanones (**2j** and **2k**), we suspect that the secondary and tertiary cyanoalkyl radicals derived from **2l-2o** are inefficiently captured by the copper catalysts compared to primary cyanoalkyl radicals. We have included these unsuccessful examples in the revised manuscript.

Comments: (3) Table 1 of optimization studies should be reformatted. Concise results from the uses of the drawn ligands should be included.

Reply: We are grateful for the reviewer's suggestions.. The results of ligand screening in Table 1 has been placed at the corresponding position.

Comments: (4) Data of ^{31}P -NMR for copper complex A and B should be included in supporting information.

Reply: We are grateful for the reviewer's comments to our work. The ^{31}P -NMR of copper complexes A and B have been added in the supplementary information (SI, p53, p55, p93, and p94).

Comments: (5) In Fig. 7, please check carefully the structures of A and B.

Reply: We are grateful for the reviewer's suggestion. We have modified the structure of A and B in the revised manuscript.

Comments: In addition, investigation of the specific LMCT step (C-D) is too preliminary to state the event of single electron transfer from N to Cu(III). How would the electron density of the quinoline ring affect this SET event as well as the performance of catalysis?

Reply: We are grateful for the reviewer's suggestion. Appropriate electron-rich ligands can enhance the reducing capability of copper catalysts in the SET events (*Acc. Chem. Res.* **1999**, *32*, 895-903; *J. Am. Chem. Soc.* **2022**, *144*, 17319-17329). In our studies, when (4-methoxyphenyl)glycinate (**1a**) was used as the substrate, only 30% yield was obtained under standard conditions. Notably, when **1e** was used, the additional coordination of N atom on quinolinyl group to the copper center enhanced the reducing capability of the Cu catalyst, which brought the significantly increase both of the product yield and stereoselectivity. In addition, the 5-methoxyquinolinyl group (**1b**) with higher density of π electron could further improve the yield and enantioselectivity of the reaction. These experiment results are consistent with the above mentioned literature reports. Considering the convenience of the removal of the direct group, we shifted the commonly used quinolinyl direct group to a more easily removable 5-methoxyquinolinyl group and further optimized the reaction conditions.

Comments: And more information about the subsequent deprotonation is highly recommended by the measurement of KIE.

Reply: We have tried several methods to deuterate glycine α -position, and all of our efforts were failed. As shown below, we placed the substrate in deuterium water and heated at 180°C for 6 h in a sealed tube. However, only N-D product was obtained in 90% yield. Using another method (patent CN108358803A), we successfully synthesized the α -D glycinate ester (**3**) in high yield. Unfortunately, when various Pd catalyst were tested, the cross coupling between **3** and 8-Br-quinoline was failed. Thus, the KIE experiments could not be performed. We speculate that the SET from N to Cu(III) and the subsequently deprotonation likely generate the α -carbon radical of glycine derivative. (*Angew. Chem. Int. Ed.* **2021**, *60*, 7669-7674)

REVIEWERS' COMMENTS

Reviewer #2 (Remarks to the Author):

I have reviewed the last version of this manuscript. I am happy to see that the authors have addressed all of my concerns and they also tried their best to solve the problems raised by other reviewers. Overall, I strongly support its publication at Nature communications.

Reviewer #3 (Remarks to the Author):

It is great to see that the revised manuscript described by Xu and coworkers has been significantly improved. Several modifications have been properly made by taking off the term of "photo-free" in Introduction, reformatting the optimization table 1, testing more advanced examples of oxime esters despite with no success and disclosing more discussions on the putative mechanism. Meanwhile, the data of characterization in Supplemental Information have been properly described and met the requisite standards for publication. Therefore, acceptance of the current version is suggested.